

# Assessing the breeding phenology of a threatened frog species using eDNA and automatic acoustic monitoring

Ying Chen*, Orianne Tournayre*, Haolun Tian and
Stephen C. Lougheed

Biology, Queen's University, Kingston, Ontario, Canada
* These authors contributed equally to this work.

Corresponding author
Stephen C. Lougheed,
lough@queensu.ca

## ABSTRACT

**Background:** Climate change has driven shifts in breeding phenology of many amphibians, causing phenological mismatches (*e.g.*, predator-prey interactions), and potentially population declines. Collecting data with high spatiotemporal sensitivity on hibernation emergence and breeding times can inform conservation best practices. However, monitoring the phenology of amphibians can be challenging because of their cryptic nature over much of their life cycle. Moreover, most salamanders and caecilians do not produce conspicuous breeding calls like frogs and toads do, presenting additional monitoring challenges.

**Methods:** In this study, we designed and evaluated the performance of an environmental DNA (eDNA) droplet digital PCR (ddPCR) assay as a non-invasive tool to assess the breeding phenology of a Western Chorus Frog population (*Pseudacris maculata* mitotype) in Eastern Ontario and compared eDNA detection patterns to hourly automatic acoustic monitoring. For two eDNA samples with strong PCR inhibition, we tested three methods to diminish the effect of inhibitors: diluting eDNA samples, adding bovine serum albumin to PCR reactions, and purifying eDNA using a commercial clean-up kit.

**Results:** We recorded the first male calling when the focal marsh was still largely frozen. Chorus frog eDNA was detected on April 6th, 6 days after acoustic monitoring revealed this first calling male, but only 2 days after males attained higher chorus activity. eDNA signals were detected at more sampling locales within the marsh and eDNA concentrations increased as more males participated in the chorus, suggesting that eDNA may be a reasonable proxy for calling assemblage size. Internal positive control revealed strong inhibition in some samples, limiting detection probability and quantification accuracy in ddPCR. We found diluting samples was the most effective in reducing inhibition and improving eDNA quantification.

**Conclusions:** Altogether, our results showed that eDNA ddPCR signals lagged behind male chorusing by a few days; thus, acoustic monitoring is preferable if the desire is to document the onset of male chorusing. However, eDNA may be an effective, non-invasive monitoring tool for amphibians that do not call and may provide a useful complement to automated acoustic recording. We found inhibition patterns were heterogeneous across time and space and we demonstrate that an internal positive control should always be included to assess inhibition for eDNA ddPCR signal interpretations.

## INTRODUCTION

Climate change is negatively impacting biodiversity and ecological processes, raising serious conservation concerns worldwide (*Walther et al., 2002*; *Walther, 2010*; *Bellard et al., 2012*; *Lister & Garcia, 2018*). Among the widely documented consequences of global warming are phenological shifts–changes in timing of seasonal activities such as migration (*Van Buskirk, Mulvihill & Leberman, 2009*), flowering (*CaraDonna, Iler & Inouye, 2014*), chorusing in frogs (*Klaus & Lougheed, 2013*), and spawning in fish (*Lynch et al., 2016*). Consequences of phenological shifts can be at the population level, such as phenological isolation from other populations and concomitantly loss of genetic connectivity (*Heard, Riskin & Flight, 2012*), but also at the individual scale (*Stillman, 2019*; *Abrahms et al., 2022*). For example, such shifts can influence reproductive success by creating a temporal mismatch between offspring and access to their preferred prey (*Visser, te Marvelde & Lof, 2012*; *Reed, Jenouvrier & Visser, 2013*), reducing successful development and fitness of individuals. Amphibians, which are among the most threatened vertebrate groups (*Stuart et al., 2004*; *Wake & Vredenburg, 2008*), exhibit the greatest changes in breeding timing as a consequence of climate change, over two times faster than trees, birds, and butterflies (*Parmesan, 2007*; *While & Uller, 2014*; *Thackeray et al., 2016*). However, amphibian species have responded heterogeneously to local climate changes (*Ficetola & Maiorano, 2016*), with some exhibiting no phenological shift (*Gibbs & Breisch, 2001*; *Klaus & Lougheed, 2013*; *Kirk, Galatowitsch & Wissinger, 2019*), some breeding earlier (*Klaus & Lougheed, 2013*; *Green, 2017*), and others breeding later (*Todd et al., 2011*; *Arnfield, Monk & Uller, 2012*; *Arietta et al., 2020*). Primary factors underlying such heterogeneity remain elusive (*Gibbs & Breisch, 2001*; *Klaus & Lougheed, 2013*). However, one reason might be that the environmental cues for amphibian breeding are species-specific (*Oseen & Wassersug, 2002*), with some associated with temperature (*Ospina et al., 2013*), rainfall (*Saenz et al., 2006*; *Ulloa et al., 2019*), photoperiod (*Schalk & Saenz, 2015*), and lunar cycle (*Grant, Chadwick & Halliday, 2009*).

Monitoring and tracking amphibian activities commonly involve invasive approaches such as drift fences, pitfall traps, and hand capture (*Heyer et al., 1994*; *Kirk, Galatowitsch & Wissinger, 2019*). However, due to the cryptic nature of most amphibians over much of their annual cycle, traditional non-invasive methods are often ineffective. For example, visual surveys for Hida salamanders (*Hynobius kimurae*) in a stream in Honshu, Japan had a detection rate of only 23.3% (*Jo et al., 2020*). Another limitation is that passive acoustic monitoring can only be used to track male anuran calling but not emergence from brumation nor female activities as they do not use advertisement vocalizations in many species. Environmental DNA (eDNA) offers a powerful alternative to assess phenology. It has been used as a non-invasive method to achieve multiple goals: (i) for detecting presence of target species; (ii) for inferring aspects of species ecology, including spawning activities in fish, based on eDNA concentrations (*Bylemans et al., 2016*; *Thalinger et al.,*

2019; *Tsuji & Shibata, 2020*); (iii) for locating turtle overwintering sites (*Feng, Bulté & Lougheed, 2019*; *Tarof et al., 2021*); (iv) for identifying frog breeding sites (*Everts et al., 2021*); and (v) for quantifying seasonal distributions of salamanders (*Jo et al., 2020*). However, challenges remain in sampling design (*i.e.*, number of samples/site) and interpretation of eDNA data, mostly because of differential rates of degradation of eDNA in the environment, different spatial ecology of species–for example patchiness of focal species, and PCR inhibition (*Beng & Corlett, 2020*). Droplet digital PCR (ddPCR) is increasingly popular in species-specific eDNA surveys because it is less prone to PCR inhibition, and is more sensitive and accurate for absolute concentration estimation compared to the more widely-used quantitative PCR (qPCR) (*Doi et al., 2015*; *Zhao et al., 2016*; *Sidstedt, Rådström & Hedman, 2020*).

The Western Chorus Frog (*Pseudacris triseriata*) is a small, well-camouflaged terrestrial treefrog species whose populations in Ontario and Québec are declining (*COSEWIC, 2008*). Taxonomy of this species is controversial largely because evolutionary relationships revealed by analyses of mitochondrial and nuclear genomes are discordant (*Lougheed et al., 2020*). The nuclear genomes appear to be similar across Ontario and Québec, but there exist two deeply diverged, non-sister mitochondrial genomes, one showing affinity to *P. triseriata* and the other most similar to *P. maculata* (the Boreal Chorus Frog) in the United States (*Lemmon et al., 2007*; *Lougheed et al., 2020*), suggesting a complex evolutionary history with hybridization and introgression. The current legal name for Ontario and Québec populations is Western Chorus Frog (*P. triseriata*) in Canada, with two Designatable Units delineated for conservation purposes (*Green, 2005*), corresponding to the two mitochondrial types (mitotypes): (i) the Carolinian population with *P. triseriatia* mitotype; and (ii) the Great Lakes/St. Lawrence—Canadian Shield population with *P. maculata* mitotype (*COSEWIC, 2008*). Western Chorus Frogs spend most of their lives foraging, resting, and brumating beneath leaf litter and woody debris in wooded areas (*Whitaker, 1971*), surviving the winter by producing a cryo-protectant that reduces probability of freezing in sub-zero temperatures (*Swanson, Graves & Koster, 1996*; *Higgins & Swanson, 2013*). Thus, they remain hidden for much of the year and become evident only in the breeding season when males call from vegetation at the water surface in marshes and vernal pools (*Whitaker, 1971*). The Western Chorus Frog is a cold breeder and is among the first to breed in spring when air temperature starts to increase, with onset usually after a few days of elevated temperatures (*Whitaker, 1971*). Male calling is associated with temperature, rainfall, and water availability in breeding habitats (*Buckley et al., 2021*). They are most vocally actively in the first half of the night, although they do call all day (*Whitaker, 1971*; *Buckley et al., 2021*). Male Western Chorus Frogs typically stay in ponds multiple days in early spring while females leave breeding ponds after ovipositing (*Whitaker, 1971*). Females release hundreds of eggs in water that, once fertilized, typically hatch in 1 to 3 weeks, depending on the water temperature (*Whitaker, 1971*). After 40–90 days, depending on temperature and hydroperiod, tadpoles metamorphose and migrate short distances into adjacent woodlands (*Ethier et al., 2021* and references therein). The phenology of Western Chorus Frog appears not to have changed between 1970 to 2010 in Eastern Ontario concomitant with regional climate change, although data are

sparse (*Klaus & Lougheed, 2013*). *Amburgey et al. (2012)* showed that *P. maculata* tadpoles grew at rates reflecting the hydroperiods of their native ponds. If climate change shortens the hydroperiod of chorus frog breeding habitat, tadpoles may have insufficient time to develop and metamorphose (*Gervasi & Foufopoulos, 2008*). In contrast, if hydroperiod is extended (*e.g.*, because of increased precipitation in early spring), water bodies may become deeper and/or permanent, resulting in increased predation risk (*e.g.*, by fish, *Skelly, 1996*) or this may impact adult oviposition behaviour and tadpole activity (*Hopey & Petranka, 1994*; *Richardson, 2001*). Western Chorus Frogs are notoriously difficult to find and catch. *Hocking et al. (2008)* did a 4-year phenology monitoring for 11 amphibian species in Missouri using drift fence and pitfall traps and Western Chorus Frog was the only species that was not caught over the entire sampling period. The most efficient way of monitoring the breeding phenology has been the passive automatic acoustic monitoring (*Buckley et al., 2021*).

In this study, we designed and validated an eDNA assay for a Western Chorus Frog population in Eastern Ontario (with the *P. maculata* mitotype)–hereafter we simply use chorus frog. We assessed the efficacy of our ddPCR assay for breeding phenology monitoring and compare these data with data from passive automatic acoustic monitoring. We also investigated three methods that have been proposed to mitigate PCR inhibition: diluting samples, adding bovine serum albumin (BSA), and purifying eDNA using a commercial kit.

## MATERIALS AND METHODS

### Study site and environmental data

Our study site (Round Field Marsh) is a shallow upland marsh encompassing approximately 8,000 m$^2$ at the Queen's University Biological Station (QUBS) in Eastern Ontario (44.5175° N, 76.3883° W) (Figs. 1 and S1). Around 50% of the marsh area comprises thickets and tall shrubs, while the rest is dominated by cattails (genus *Typha*; Fig. 1). From March 8$^{th}$ to July 25$^{th}$, 2022 we recorded hourly humidity and air temperature using a HOBO MX2303 data logger (Onset Computer Corporation, Bourne, MA, USA), and water temperature using a HOBO UA-001-08 Pendant data logger (Onset Computer Corporation, Bourne, MA, US). We obtained precipitation and wind speed data from an AcuRite Atlas weather station (Chaney Instrument Co., Lake Geneva, WI, USA) located at 44.585° N, 76.304° W, 10 km away from the study site. A DJI Mavic 2 Air (DJI, Shenzhen, China) unmanned aerial vehicle (UAV) was deployed at the field site during each eDNA sampling session to characterize the field site including ice/snow cover.

### Automated acoustic monitoring

We placed three Wildlife Acoustics Song Meters SM4 acoustic recorders on March 8$^{th}$, 2022, when the marsh was still frozen and covered with snow (Fig. 1). Song Meters were programmed to record the first 3 min each hour. We retrieved the data after the chorus frog breeding season (July 2022) and found one recorder had experienced technical issues with discontinuous recordings and many gaps (Song Meter C; Fig. 1C). We thus primarily used data from two recorders (Figs. 1A and 1b) with the third (Fig. 1C) for supplementary

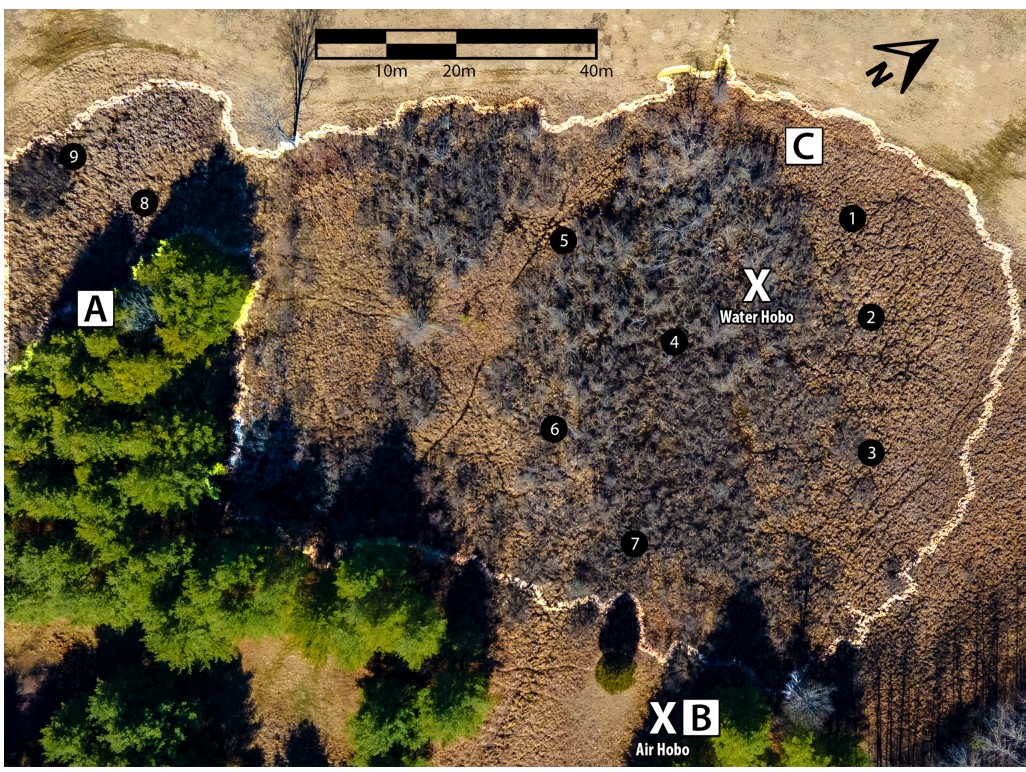

**Figure 1 Locations of eDNA samples, Wildlife Acoustic Song Meter recorders and temperature Hobo loggers in Round Field Marsh.** Locations of three Wildlife Acoustics Song Meter recorders (A, B and C in squares), air and water temperature Hobo loggers and nine eDNA sampling locales (in circles) at Round Field Marsh. The marsh was heterogeneous with patches of thickets and cattail reeds. The approximate boundaries of the marsh are highlighted in white.

insights if needed. We listened to and visually inspected the spectrograms in Kaleidoscope Pro 5.4.6 (Wildlife Acoustics Inc, Maynard, MA, USA) and estimated the number of calling males. Chorus frogs have distinctive pulsed advertisement calls with dominant frequency at around 3.5 kHz (Fig. S2; *Bee et al., 2010*; *Nityananda, Bee & Coleman, 2011*). Inter-individual call variation allowed counting individuals both audibly and visually and up to five individuals were distinguishable with confidence (Fig. S2). If we estimated different numbers of calling males from the two recorders (due to background noise and/or the distance of calling males to the recorders), we used the higher of the two values.

We tested which environmental variables related to chorus frog calling intensity at our study site. We classified calling activity as binary for each hour (1: calling, 0: non-calling) from 0AM on March 31st (first day of male call), to 11PM on April 12th, the day that eDNA sampling ended. We explored the association between environmental variables and hourly calling activity using a Generalized Linear Model (GLM) with a quasibinomial family (over-dispersion) in R version 4.0.5 (*R Core Team, 2021*). Water temperature (°C), relative humidity (%), precipitation rate (mm/h), wind speed (km/h), and calling in the previous hour (1: calling, 0: non-calling) were included as predictor variables. Adding calling in the previous hour as another predictor variable allowed us to (partially) account for

non-independence of hourly calling activity. We did not include air temperature in the model because air temperature was strongly correlated with water temperature (Spearman's correlation = 0.7; *ggcor* function of the GGally package) and male chorus frogs typically call at the water surface.

## eDNA assay design and validation

We designed a Taqman probe-based eDNA assay for chorus frog (*P. maculata* mitotype) for the cytochrome *b* (cyt*b*) mitochondrial gene in Geneious v7.1.9 (*Kearse et al., 2012*) using parameter values recommended by *Klymus et al. (2020)* (see Table S1 for details). We used 100 sequences produced by *Lougheed et al. (2020)* and 124 sequences available in GenBank for target and co-occurring species (Table S2). The forward (5′-CGTAGCA CACATCTGCCGTG-3′) and reverse primers (5′- CCGACGAAGGCTGTTGCTAT -3′) labeled with the double-quenched probe (5′ 6-FAM- ACGCAACCTGCACGCAAACG GA-Iowa Black FQ -3′) amplified a 196 bp amplicon, with at least nine mismatches with any of the nine co-occurring anuran species and five mismatches with the *P. triseriata* cyt*b* mitotype sequence in Ontario (Tables S2 and S3). While other assays exist for chorus frogs in Alberta (*Booker, 2016*), Ontario (*Beauclerc et al., 2019*), and Québec (*Hernandez et al., 2020*; *Dubois-Gagnon et al., 2022*), we designed an assay that is effective for ddPCR and that could reliably distinguish the two chorus frog mitotypes in Ontario from other co-occurring anuran species in the region.

   Optimal annealing temperature was 65 °C based on an initial thermal gradient experiment testing temperatures from 62.8 °C to 68.1 °C on extracted chorus frog DNA (*P. maculata* mitotype) on a Bio-Rad QX200™ AutoDG™ Droplet Digital™ PCR System (ddPCR) (Fig. S3). We assessed the specificity of the ddPCR assay on co-occurring species and the *P. triseriata* mitotype from Ontario. We prepared four replicates of each sample using the same ddPCR mix and PCR conditions, three for ddPCR detections and one for Sanger sequencing. The PCR products from the fourth replicate were purified and Sanger sequenced at Génome Québec (Montréal, Canada). None of the co-occurring species amplified, and Sanger sequencing revealed that amplifications of the *P. triseriata* mitotype with our *P. maculata* primers were due to cross contamination (Fig. S4). Finally, we validated the assay using eDNA water samples collected in 2021 (triplicates for each site) from: (1) four local marshes where chorus frogs are known to be present, and (2) five marshes and two lakes where chorus frogs do not occur (Table S4). All the amplicons from 2021 were confirmed to be the correct species through Sanger DNA sequencing following the same protocol as above.

   We used an 11-point 1:2 dilution series of the chorus frog (*P. maculata* mitotype) synthetic DNA (Integrated DNA Technologies gBlocks™, 216 bp) with six replicates to assess the Limit of Detection (LOD) and Limit of Quantification (LOQ) of the assay. The highest concentration of this series, measured with ddPCR, was 4.64 copies/µL in the reaction mix. We define the LOD as the highest concentration below which false negatives may be expected (*Klymus et al., 2020*), and this is the lowest standard concentration at which the chorus frog synthetic DNA could be detected (*i.e.*, at least one positive droplet in one replicate) (*Brys et al., 2021*). If a positive signal was detected in a control sample

(*i.e.*, field control, filtration control, extraction control or ddPCR no-template control; all used to detect false positives), LOD was set to three times the number of positive droplets observed in the control sample to account for possible false positive detections from cross-contamination. LOQ was defined as the lowest standard concentration of the linear range of the standard curve, before the measured concentrations plateaued (*Brys et al., 2021*).

## Water sampling, filtration and eDNA extraction

We sampled water every 2 days starting on March 23$^{rd}$, 2022 when Round Field Marsh began to thaw (*i.e.*, some patches of open water). No samples were collected on March 29$^{th}$ and 31$^{st}$ when the surface of Round Field Marsh re-froze due to cold weather (Fig. S5). We stopped sampling on April 12$^{th}$, 2022 after we heard more than three frogs calling during sampling. We collected between three and nine 1L samples of surface water depending on the amount of ice-free open water using sterile 1L bottles (Nalgene HDPE Narrow Mouth) (Fig. 1). For each sampling day, we included a field negative control (bottle filled with distilled water left open during sampling). The sampling bottles were stored individually in Ziploc$^{TM}$ bags in a cooler with ice packs for return to the lab for filtration. All equipment (including waders) was decontaminated with 10% bleach solution and rinsed with distilled water before and after sampling. Sterile nitrile gloves were changed between water samples.

We filtered the water samples within 6 h of sampling in a dedicated room using Polycarbonate (PCTE) membrane filters (pore size: 1.0 µm, diameter: 47 mm; Sterilitech) housed in a 47 mm in-line filter holder and a Waterra portable peristaltic pump. Prior to filtration, the filter holder and pump tubing were submerged in 10% bleach overnight and thoroughly rinsed using distilled water. The filter holder and tubing were also bleached with 1L of 10% bleach and rinsed with 1L of distilled water between each liter of water sampled. After filtration, the filters were folded with tweezers (dipped in 95% ethanol and flamed to disinfect) and stored in 2 mL tubes containing 700 µL 2% (w/v) cetrimonium bromide extraction buffer (CTAB; *Turner et al., 2014*). All filters were immediately stored at −20 °C until DNA extraction. When several filters were used for the same 1L sample, each filter paper was individually stored in 700 µL CTAB for independent extraction. We assessed the possibility of cross-contamination by filtering 1L distilled water as a filtration negative control.

The eDNA extractions were conducted using a modified chloroform-based method from *Turner et al. (2014)* by repeating the steps of adding equal volume of 24:1 chloroform: isoamyl ethanol and skipping the step of adding 5 M NaCl. We eluted the DNA in a final volume of 50 µL nuclease free water. For 24 water samples with multiple filters (two to six filters), we applied a serial elution method to maximize yield. We added 25 µL of nuclease free water to the first filter tube and incubated at 56 °C for 5 min. We then transferred the 25 µL eluate from the first filter tube to the second filter tube and incubated at 56 °C for 5 min. This was repeated for all sequential filter tubes. We conducted the serial elution for each sample twice, so that the 25 µL of eluate from the first and second serial elution added up to 50 µL. We included one no-template control (NTC) for each DNA extraction session (three in total).

## eDNA detection and quantification in ddPCR

All samples were run in triplicate with NTCs and chorus frog genomic DNA positive controls (PCs) included in each ddPCR run to test for false negatives and positives, respectively. The PCR mastermix was made to a volume of 25 μL: 12.5 μL of Bio-Rad ddPCR Probe no dUTP mix (2X), 2.25 μL of forward primer (10 μM), 2.25 μL of reverse primer (10 μM), 0.625 μL of Taqman probe (10 μM), 0.175 μL of nuclease free water and 7.2 μL of DNA. PCR cycling conditions were as follows: 95 °C for 10 min, 45 cycles of 94 °C for 30 s and 65 °C for 1 min, then 98 °C for 10 min and holding at 4 °C. We used the direct quantification mode in the QX Manager 1.2 Standard Edition software (BioRad, Hercules, CA, USA). The fluorescence threshold was set for each plate (*Baker et al., 2018*; BioRad Droplet Digital™ PCR Application Guide) using the corresponding NTCs and PCs as reference for positive and negative droplets. Concentrations were reported as number of copies/μL (*i.e.*, ddPCR concentration measurement) and number of copies in the original sample (1L of water, copies/L) calculated as follows:

$$\frac{\left(\dfrac{ddPCR\ concentration\ \times\ Volume\ ddPCR\ mix}{Volume\ of\ DNA\ in\ the\ ddPCR\ mix}\right) \times Volume\ of\ DNA\ extract}{Volume\ of\ filtered\ water}$$

## Inhibition estimation and mitigation strategy

We used a northern map turtle (*Graptemys geographica*) probe assay (*Feng, Bulté & Lougheed, 2019*) as an internal positive control (the northern map turtle does not inhabit shallow marshes) to test for inhibition in all eDNA samples in a separate ddPCR run (*i.e.*, no multiplexing). The PCR mastermix was made to a volume of 25 μL: 12.5 μL of Bio-Rad ddPCR Probe no dUTP mix (2X), 1.125 μL of *G. geographica* forward primer (20 μM), 1.125 μL of *G. geographica* reverse primer (20 μM), 0.625 μL of *G. geographica* Taqman probe (10 μM), 0.425 μL of nuclease free water, 2 μL of *G. geographica* synthetic DNA and 7.2 μL of eDNA samples (or nuclease free water for the no-template controls). PCR cycling conditions were adapted from *Feng, Bulté & Lougheed (2019)*: 95 °C for 10 min, 45 cycles of 94 °C for 30 s and 60 °C for 1 min, then 98 °C for 10 min with a final holding step at 4 °C. All eDNA samples were run in one replicate.

As we observed strong inhibition in some samples, we tested three mitigation methods on two of them (10L9, 12L5): (1) dilution of the eDNA samples to 1:5 and 1:10 ratio with sterile nuclease free water, (2) addition of bovine serum albumin (BSA; 20 mg/mL) to the ddPCR reaction mix (see below) of the 1:10 diluted samples, and (3) purification of the undiluted eDNA samples using the DNA Clean-Up and Concentration Micro Elute kit (Norgen Biotek Corp., Ontario, Canada) which were eluted in a final volume of 20 μL (2X 10 μL). We also purified two eDNA samples with high concentrations of chorus frog eDNA from 2021 from two local marshes in Kingston, Ontario (Cecil Graham Park and Old Mill Road sites, Table S4) to estimate the target eDNA recovery rate from the clean-up kit. We used the same PCR mastermix recipe for the 1st and 3rd methods (see the eDNA detection and quantification in ddPCR section). We used the following recipe for the 2nd method: 12.5 μL of Bio-Rad ddPCR Probe no dUTP mix (2X), 1.125 μL of chorus frog

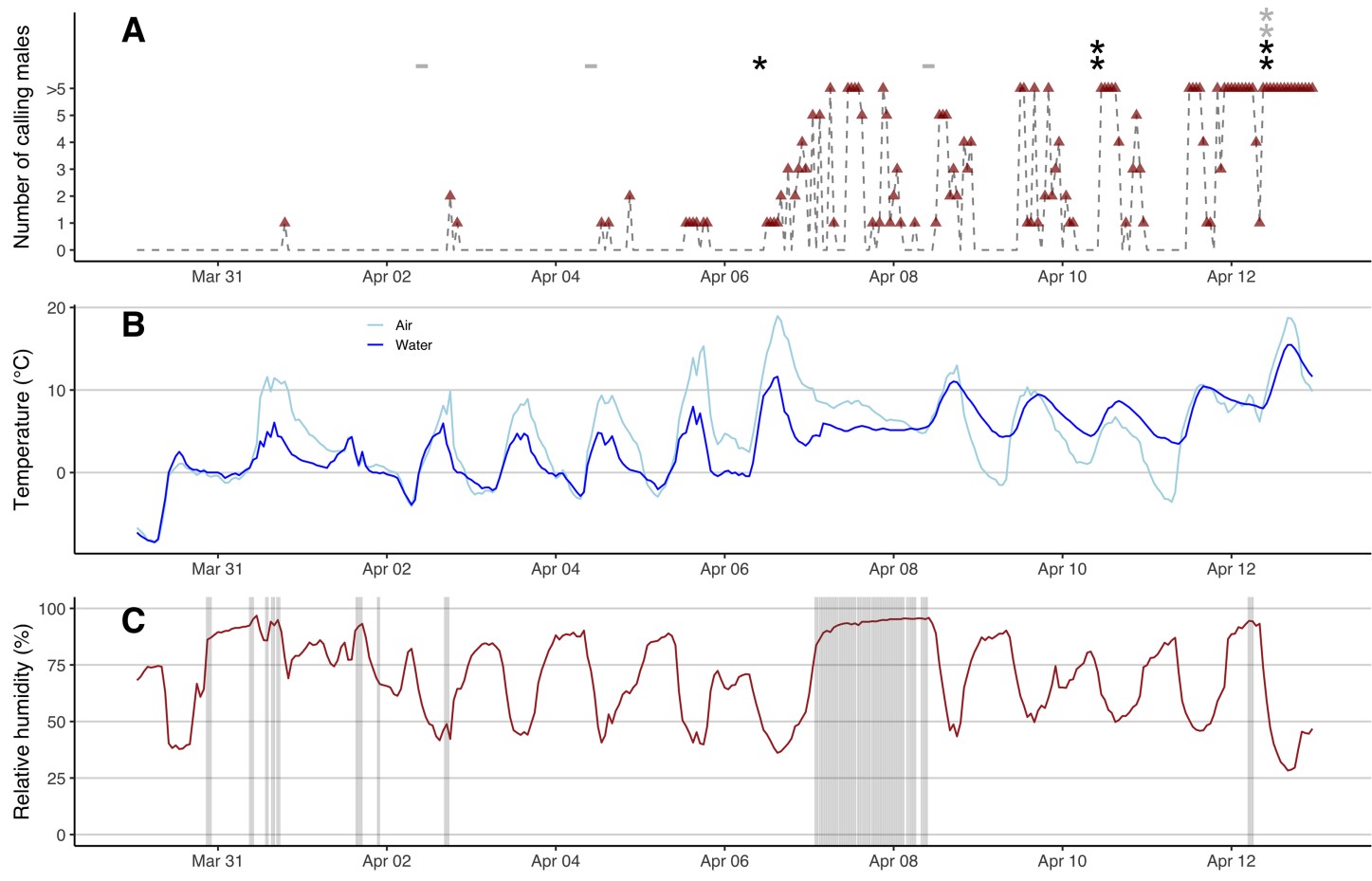

**Figure 2** **eDNA detections and hourly chorus frog calling activities, air and water temperatures, precipitation and relative humidity.** (A) Hourly number of calling males (red triangles, grey dash lines) from 0 AM on March 30th to 11 PM on April 12th, 2022. Black stars represent positive eDNA detections and grey stars represent inconclusive eDNA detections. Grey minus symbols represent negative eDNA detections. (B) Hourly air and water temperature (°C). (C) Hourly relative humidity (%) shaded with precipitation.

forward primer (20 µM), 1.125 µL of chorus frog reverse primer (20 µM), 0.625 µL of chorus frog Taqman probe (10 µM), 1.925 µL of nuclease free water, 0.5 µL BSA and 7.2 µL of eDNA or nuclease free water for no-template controls. All samples were run in duplicate.

# RESULTS

## Male calling activities and environmental conditions

We first found open water patches on March 23rd under consistent above-zero temperatures, but these refroze completely during a cold snap (air temperature down to −10 °C) from March 27th to March 30th (Fig. S5). The marsh started to re-thaw on March 31st when the daily maximum air temperature exceeded 5 °C. After a warm rain in the afternoon of March 31st, the first calling male was heard at 7 PM (Fig. 2A), despite the fact that over 90% of the marsh was covered with ice that morning (Fig. S5). Calls were evident from recordings from Song Meter B but not A (the data were missing in recordings from

C), suggesting that the male called near the east side of the marsh where the ice first started to thaw (Figs. 1 and S5). No additional calling was heard the rest of day nor on April 1$^{st}$. After a warm rain in the afternoon of April 2$^{nd}$, two males were heard at 6 PM from both Song Meter B and C (but not A) with clearer sonograms evident in C recordings, suggesting that males were closer to C in the east side of the marsh (Fig. 1). One male was heard calling at 8 PM in recordings from all three recorders (Fig. 2A). The number of hours that males called was low during these first few days and males called only briefly on April 4$^{th}$ and April 5$^{th}$ (Fig. 2A). On the warm, rainy day of April 6$^{th}$, when the maximum air temperature was 18.9 °C, many males emerged and started to call (Fig. 2). The calls were louder and clearer in sonograms from Song Meter A with fewer or even no individuals heard from Song Meter B, suggesting many individuals emerged near the west side of the marsh (Fig. 1). The marsh was mostly thawed by April 8$^{th}$ (Fig. S5) and the water levels were high due to the rains on April 7$^{th}$ and 8$^{th}$ (Fig. 2C). Male calling activities were intermittent despite the presence of many males in the marsh between April 8$^{th}$ and April 11$^{th}$. Beginning on April 12$^{th}$, the chorus remained continuous. Calling activity (*i.e.*, number of hour-time slots for which calling was recorded) in the early breeding season increased with warmer water temperatures (partial R$^2$ = 0.248, F$_{1,308}$ = 72.474, $p$ < 0.001) and was not affected by humidity ($p$ = 0.801), precipitation ($p$ = 0.236), nor wind speed ($p$ = 0.067). Calling in the previous hour was also a significant predictor (partial R$^2$ = 0.059, F$_{1,309}$ = 128.998, $p$ < 0.001).

## Sensitivity of the ddPCR assay and eDNA detections

The LOQ was set as 0.11 copies/µL (the mean concentration of the three positive 1/32 dilution replicates) (Fig. 3). Any eDNA samples with a concentration below 0.11 copies/µL were considered true detections but could not be accurately quantified. The raw reading of the LOD was 0.06 copies/uL (one positive replicate) but because this concentration is below the LOQ, we calculated the LOD (1/512 dilution) by dividing the LOQ (*i.e.*, 1/32 dilution) by 16, which resulted in 0.007 copies/µL.

In total, 56 eDNA samples were collected and run in triplicate using ddPCR: two samples on March 23$^{rd}$, four on March 25$^{th}$, four on March 27$^{th}$, six on April 2$^{nd}$, six on April 4$^{th}$, seven on April 6$^{th}$, and nine on April 8$^{th}$, 10$^{th}$ and 12$^{th}$ respectively (Fig. 4). In the ddPCR runs, one replicate from locale 7 sample taken on April 6$^{th}$ and two replicates from locale 5 taken on April 10$^{th}$ did not generate sufficient droplets for valid interpretation (*i.e.*, fewer than 10,000, potentially because of inhibitors, see below). Those replicates were excluded from subsequent analysis. The total number of generated droplets across all other samples and their replicates was on average 19,216, with a minimum of 10,015 droplets and a maximum of 21,758. All field, filtration, extraction and ddPCR controls were negative, except one field control replicate from April 12$^{th}$, which had two positive droplets; thus, we considered eDNA samples from April 12$^{th}$ positive only if the number of positive droplets was higher than six.

The earliest positive eDNA sample was from locale 5 on April 6$^{th}$ with only one positive replicate (Figs. 4 and 5E) although the concentration could not be estimated with confidence (<LOQ). No chorus frog eDNA was detected on April 8$^{th}$, despite the presence

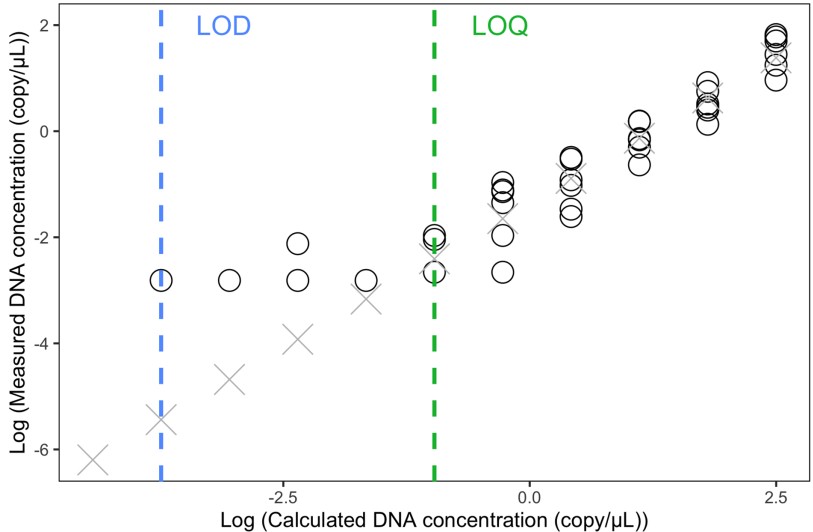

**Figure 3 Limit of Detection and Limit of Quantification of the eDNA ddPCR assay.** Estimation of Limit of Detection (LOD) and Limit of Detection (LOQ) using 1:2 dilution series with 11 steps (1, 1/2, 1/4, 1/8, …, to 1/1,024 dilution). Each concentration was run with six replicates in ddPCR. Circles indicated replicates with positive eDNA signals. Crosses were predicted concentrations based on the linear model of the five highest concentrations.

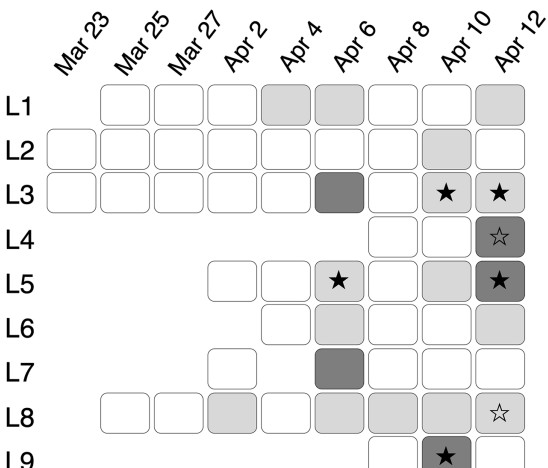

**Figure 4 eDNA signal and inhibition patterns across sampling days.** eDNA samples collected from March 23rd to April 12th at locale 1 (L1) to locale 9 (L9). Positive eDNA detection is indicated with a filled star. Inconclusive eDNA detection is indicated with an open star. A dark grey background indicates strong inhibition in the eDNA sample as revealed by the internal positive control. A light grey background indicates minor inhibition as revealed by the internal positive control.

of calling males in the marsh (Fig. 2A). Two eDNA samples on April 10th showed positive signals (10L9 and 10L3; Figs. 4, 5A and S6A). 10L3 had one positive replicate although again concentration could not be estimated with confidence (<LOQ, Fig. S6A). All three technical replicates from 10L9 showed positive signals with a mean concentration at 1.28 copies/µL (222.5 copies/L in 1L sample), which was underestimated due to PCR

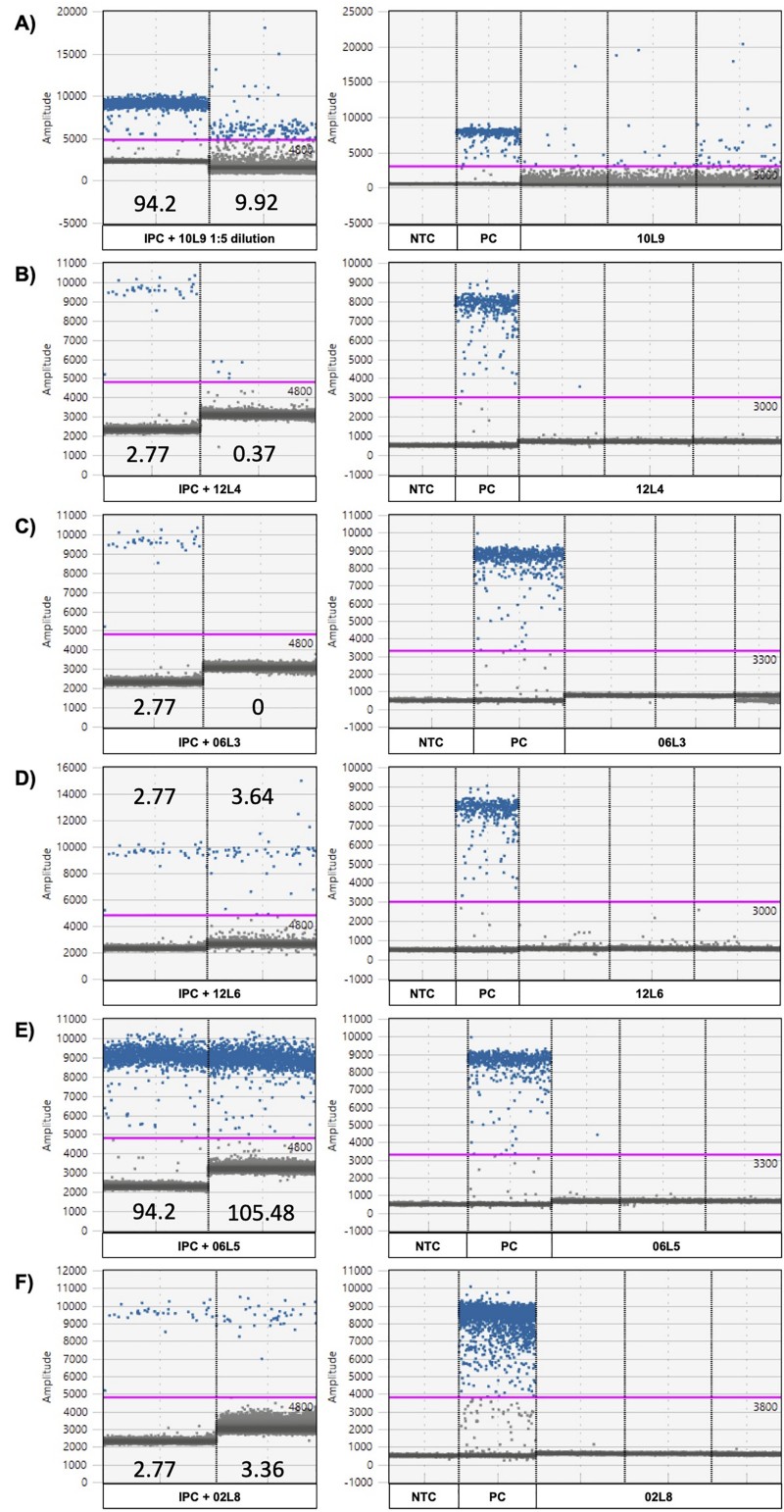

**Figure 5 ddPCR amplitude graphs of eDNA samples showing strong or minor inhibition.** Strong (A–C) and minor inhibitions (D–E) revealed by internal positive control (IPC positive control and IPC spiked in eDNA sample; left) and ddPCR amplitude graphs of the chorus frog (*Pseudacris maculata* mitotype) target assay (right). 10L9 1:5 dilution = locale 9 on April 10th with a 1:5 dilution, 10L9 = locale 9

**Figure 5** (continued)
on April 10[th], 12L4 = locale 4 on April 12[th], 06R5 = locale 5 on April 6[th], 12L6 = locale 6 on April 12[th], PC = positive control using *P. maculata* genomic DNA and NTC = no template control. The numbers in the IPC panel are the concentrations of the IPC as read by the ddPCR instrument (copies/μL). The pink line represents the threshold for positive (blue) and negative (gray) droplets, which is set based on positive control and no-template control in each plate.

inhibition (Fig. 5A). Two samples on April 12[th] were also positive for all three replicates (Fig. 4), locale 3 at 0.36 copies/μL (62.7 copies/L, Fig. S6B) and locale 5 at 2.48 copies/μL (431.2 copies/L, although the concentration was underestimated due to PCR inhibition, Fig. S6C). Finally, two samples (locales 4 and 8 on April 12[th]) were inconclusive because the number of positive droplets was lower than six (*i.e.*, lower than the number of positive droplets in the no-template control; Figs. 5B and S6D). In summary, one sample had positive eDNA detections on April 6[th], two on April 10[th], and two on April 12[th] with two additional inconclusive samples (Fig. 4).

## PCR inhibition

Using the *G. geographica* IPC, all samples generated enough droplets to be interpreted with confidence (>10,000 droplets). Five samples (April 6[th] L3 and L7, April 10[th] L9, April 12[th] L4 and L5; Fig. 4) showed strong inhibition with the collapse of positive droplets that either lowered the concentration estimates or resulted in false negative detections (Figs. 5A–5C) or a higher amplitude of some positive droplets (10L9 and 12L5; Figs. 5A and S6C). Two samples with positive chorus frog detection (10L9, 12L5) and one with inconclusive detection (12L4) were strongly inhibited (Fig. 4).

We also identified 15 samples showing minor inhibition in the IPC assay (April 2[nd] L8, April 4[th] L1, April 6[th] L1, L5, L6 and L8, April 8[th] L8, April 10[th] L2, L3, L5 and L8, April 12[th] L1, L3, L6, L8; Fig. 4). Minor inhibition did not affect positive detections but can increase concentration estimations (over 20% variation of the IPC concentration in the NTC; *Shehata et al., 2017*) (Figs. 5D and 5F). They showed one or more of the following patterns: (1) elevated fluorescence of some positive droplets (Fig. 5D), (2) elevated fluorescence amplitude of the negative droplet baseline (Figs. 5E and 5F), and/or (3) broader width of the negative droplet baseline (Fig. 5F). Those minor inhibition patterns were not always observed in the corresponding target chorus frog assay. Three samples with positive chorus frog detections (6L5, 10L3, 12L3) and one that was inconclusive (12L8) showed minor inhibition in the IPC assay (Figs. 5 and S6).

Of the three mitigation methods used to assess inhibition of the locale 9 sample from April 10[th] (10L9) and the locale 5 sample from April 12[th] (12L5), dilution was the most effective (Fig. S7). The 1:5 and 1:10 dilutions of 10L9 had mean concentrations of 1.5 copies/μL and 0.55 copies/μL respectively, corresponding to 1,298 copies/L and 948 copies/L in the original 1L sample. The 1:5 and 1:10 dilutions of 12L5 had mean concentrations of 2.765 copies/μL and 0.615 copies/μL respectively, corresponding to 2,402 copies/L and 1,074 copies/L in the original 1L sample. With dilution, the estimated chorus frog eDNA concentrations were approximately four to six times higher in sample 10L9 and two to five times higher in sample 12L5 (Fig. S8). However, the strong inhibition patterns

were still evident because the collapse of positive droplets remained in the diluted 10L9 and 12L5 samples, resulting in lower concentration estimates (Fig. S8). The combination of the 1:10 dilution with BSA generated either lower or no eDNA positive signals in both samples (Figs. S7 and S8). The commercial clean-up kit did not generate any positive eDNA signal in either of the two samples, possibly due to target eDNA being washed away and/or the inhibitors not removed completely (Figs. S7 and S8). The estimated target eDNA recovery rate of the clean-up kit was 61.14% in the Cecil Graham Park sample and 85.43% in Old Mill Road sample (Fig. S8; these two samples were not inhibited and were collected in 2021 for eDNA assay validation).

## DISCUSSION

### Assay design and validation

We developed and comprehensively validated a novel ddPCR probe assay for chorus frog (*P. maculata* mitotype) detection in wetlands in Eastern Ontario. Species-specific detection can be challenging when attempting to distinguish between closely related species, subspecies, or mitotypes (*Gorički et al., 2017*; *Boardman et al., 2021*). Insufficiently specific primers and probes between closely related species or mitotypes might result in both false positive (amplification of non-target species) and negative (competition between target and non-target DNA) detections (*Wilcox et al., 2013*). Our assay was rigorously designed and tested using *in silico* approaches, *in vitro* tests with *P. maculata* genomic DNA and genomic DNA from co-occurring anuran species plus *P. triseriata*, as well as *in situ* tests on positive eDNA samples (known presence of *P. maculata* in the water bodies), and negative eDNA samples (known absence of *P. maculata* in the water bodies). LOD and LOQ were estimated, and all positive amplicons generated during the validation process were Sanger sequenced (substantial validation at level 4 out 5 on the eDNA assay validation scale from *Thalinger et al., 2021*). Following further testing of specificity on samples from a chorus frog *P. triseriata*/*P. maculata* mitotype contact zone in Southern Ontario, our ddPCR also could be used to improve the geographic delimitation of the two Designable Units currently recognized in Ontario (*COSEWIC, 2008*).

### The promise of eDNA for assessing breeding phenology of chorus frogs

Both acoustic monitoring and eDNA surveys showed that chorus frog started breeding in early spring when air temperature was low and the marsh just started thawing (Fig. 2; *Whitaker, 1971*). The ddPCR eDNA approach was effective in detecting chorus frogs at low abundance in a shallow marsh even when only a few males were calling. The first eDNA detection was on April 6th, 6 days after the first male called but only 2 days after males attained higher chorus activity. This temporal lag was not unexpected; the first male called as soon as the marsh started to thaw and one would expect that it would take time for sufficient amounts of DNA to be shed into the water. Moreover, eDNA was detected at more locales and the eDNA concentrations at locale 3 and 5 increased with more calling activities in later days, suggesting that overall eDNA may be a reasonable proxy for calling assemblage size and location and that the detection of *P. maculata* mitotype was initially
limited by the species small body mass (<1 g), its behavior (*i.e.*, sit on floating vegetation while calling) and a lower number of active individuals in the water. Frogs and salamanders with higher body mass and more mobility within breeding water bodies would likely have higher eDNA detection rates using ddPCR (*Pilliod et al., 2014*; *Goldberg et al., 2016*; *Goldberg, Strickler & Fremier, 2018*). Although eDNA revealed the onset of male activities a few days later than acoustic surveys, our results imply that, with sufficient spatial sampling coverage, eDNA may be an effective approach for detecting chorus frogs before oviposition and emergence of tadpoles (*Bylemans et al., 2016*; *Buxton et al., 2017*; *Dunn et al., 2017*; *Everts et al., 2021*). By extension, eDNA might be an effective means for broad-scale surveys of other anurans that do not vocalize (*e.g.*, newts), or whose calls are barely perceivable (*e.g.*, African clawed frog).

Our eDNA data also revealed interesting fine spatial patterns of chorus frogs that may be difficult to determine using acoustic monitoring, especially with one acoustic recorder, wind, background noise, or overlap of male calls. For example, on the east side of the marsh (locales 3 and 5) we detected chorus frogs eDNA twice while at locales 1, 2, 6 and 7 we never detected chorus frogs. This heterogeneity within the site could be explained by either a difference in the number of active chorus frogs (*i.e.*, more eDNA was shed in some locales and not others), or differences in water physicochemical factors that allowed longer/shorter persistence of eDNA.

Intensive spatial and temporal eDNA sampling is necessary for effective species surveys when abundances are low (*Goldberg et al., 2016*; *Goldberg, Strickler & Fremier, 2018*). We collected multiple 1L water samples every 2 days with the number of samples roughly proportional to the marsh area that was free of ice. The percentage of positive samples was less than 25% on April 6th and 10th but increased to almost 50% on April 12th when many males were present and calling day and night. Low biomass of the focal taxon and the lentic wetland properties with natural barriers of dense vegetation separating water patches, reducing dispersion of shed eDNA, might be the important factors underlying the lower detection rates earlier in the breeding season (*Goldberg, Strickler & Fremier, 2018*; *Shackleton et al., 2019*). These factors should be considered in eDNA sampling design; for example, conducting a pilot study can be useful to refine sampling schemes (*Goldberg et al., 2016*). In addition, climate variables are important in eDNA surveys and data interpretation. For example, the breeding activities in some anuran species are triggered by rainfall (*Saenz et al., 2006*; *Ulloa et al., 2019*) but our data imply that rainfall can dilute eDNA and reduce detection probabilities. Thus, repeated sampling across time and collection of abiotic variables such as rainfall are important for interpreting eDNA results (*Buxton et al., 2017*; *Akre et al., 2019*).

## Inhibition in ddPCR

PCR inhibition can be a challenge for the detection and quantification of eDNA (*Sidstedt, Rådström & Hedman, 2020*), especially in shallow wetlands with potentially high saprotrophic activity, low pH, and proximity to sediments that can increase the concentration of PCR inhibitor substances (*Harper et al., 2019*). In water samples, inhibitors can be organic (debris, fulmic and humic acids, phenol) or metallic (metal
ions) (*Kim et al., 1990*; *Abbaszadegan et al., 1993*; *Schrader et al., 2012*). Several inhibition mechanisms can co-occur in PCR reactions (*Opel, Chung & McCord, 2010*; *Alaeddini, 2012*; *Schrader et al., 2012*): (i) interaction with the polymerase limiting PCR efficiency (*e.g.*, lowering of the rate of extension); (ii) competition between inhibitors and reagents; (iii) interaction with the DNA (*e.g.*, degradation); and (iv) interaction with the fluorescence dye (*e.g.*, a quenching effect). While we used a small elution volume (50 μL) which could have concentrated inhibitors and DNA, we limited the presence of inhibitors by processing our samples using current best practices (multiple filters, CTAB-based storage method, and chloroform-isoamyl extraction; *Hunter et al., 2019*) and using a ddPCR approach, which is less sensitive to PCR inhibitors than qPCR (*Verhaegen et al., 2016*).

Nevertheless, our internal positive control revealed strong inhibitions in five samples that caused lower concentration estimations and potential false negatives. Minor inhibition was observed in 15 samples that did not seem to affect detection but sometimes increased concentration estimation, although this might be assay specific as the patterns were not always consistent between IPC and our target chorus frog assay. There was no obvious spatial pattern of inhibition in the marsh (Fig. 4). We found less inhibition when ice was thawing and after rain suggesting a potential dilution effect, followed by more inhibition when the marsh was fully thawed which could be explained by the release of humic substances. Warmer water temperatures could potentially activate humification (plant decay and transformation by microbial community) which would increase concentrations of humic substances in the water. Although ddPCR can be robust to high concentrations of humic acids (up to 1,000 ng/μL; *Kolar, 2015*), the effects of other humic compounds such as fulvic acids or humins have not been evaluated. We observed patterns of inhibition similar to *Maheshwari et al. (2017)* (citrus leaf petiole and fruit columella extract), *Zhao et al. (2016)* (copper-containing bactericides), and *Kolar (2015)* (metal ions). We did not investigate the source of inhibition nor the underlying mechanisms affecting eDNA detection probabilities in lentic systems, but metals and bactericides are unlikely to be the main source of inhibition in our study as one would expect that metal concentration from natural sources would be spatially homogeneous in the marsh and would not fluctuate daily (Fig. 4); moreover, there is no agricultural treatment around the site. We found samples with minor inhibition patterns sometimes had higher IPC concentrations (over 20% variation of the IPC concentration in the NTC; *Shehata et al., 2017*), while samples with strong inhibition always had lower IPC concentrations. This could be due to different concentrations of inhibitors in the sample causing different effects. For example, *Zhao et al. (2016)* found that copper had an enhancing effect at low concentration levels on both ddPCR and qPCR but an inhibitory effect at high levels. Some locales showed consistent minor inhibition (L8), with an increase or a decrease of inhibition intensity over time (*e.g.*, L4, L5 and L3, L7, respectively; Fig. 4). However, after the heavy rains on April 7th and 8th, we detected no inhibition in any of the locales save for L8 (minor inhibition). The rain, by increasing water level/volume, could have diluted inhibitors and eDNA. In sum, quantification and detection of chorus frogs (*P. maculata* mitotype) in our study may have been impeded by inhibition issues, especially early in the season when eDNA was at low

abundance in the marsh. Our results highlight the value of IPCs in eDNA studies, even when using a ddPCR approach. Although well established in qPCR studies, the use of an internal positive control to detect inhibition in environmental samples is still rare in ddPCR studies (but see *Shehata et al., 2017*; *Everts et al., 2021*).

Common solutions to PCR inhibition include adding BSA that can bind inhibitory molecules, diluting the samples, and purifying the DNA using commercial clean-up kits (*Opel, Chung & McCord, 2010*; *Goldberg et al., 2016*; *Sidstedt, Rådström & Hedman, 2020*; *Takasaki et al., 2021*). We found that diluting samples was the most effective in mitigating inhibition (*i.e.*, increased eDNA concentrations), but the collapse of droplets was still evident. The drawback of dilution is that it can also dilute the target eDNA concentration to below the Limit of Quantification (LOQ) or Limit of Detection (LOD) and so may only work for samples with higher eDNA concentrations. Although BSA has been found to be effective in qPCR for some inhibitors (*Sidstedt et al., 2015*), it decreased the eDNA signals in our ddPCR tests. Finally, purification using a commercial clean-up kit resulted in varying eDNA recovery rates among samples (*McKee, Spear & Pierson, 2015*) and no eDNA detections in the two samples, likely due to the complete loss of target eDNA. In summary, the clean-up kit and diluting samples can reduce inhibition, but can also reduce eDNA concentration below LOD, and one should interpret such results with caution especially as these might result in false negative signals (*McKee, Spear & Pierson, 2015*; *Mauvisseau et al., 2019*).

## CONCLUSIONS

We designed and thoroughly validated a sensitive eDNA-based ddPCR species detection assay, that can accurately detect a specific mitotype of the Western Chorus Frog (*P. maculata* mitotype). This will be particularly useful to study the contact zone of *P. maculata* and *P. triseriata* mitotypes in Southern Ontario. eDNA-based ddPCR approach can be used to quantify the spatial distributions of individuals within a marsh and overall may be a good proxy for frog calling assemblage size, although acoustic monitoring remains the preferred method to assess early spring breeding activities for species that vocalize. PCR inhibition remained a challenge for eDNA detection and quantification even in ddPCR studies and including an internal positive control was necessary to interpret the data, especially when the abundance of the target species was low. Finally, we found diluting extracted eDNA samples was the best in mitigating the effects from inhibitors, further testing on a higher number of samples will provide more insights on best mitigation methods. Altogether, our results show the promise of eDNA as an effective, non-invasive tool for assessing breeding phenology in non-calling amphibians and the importance of using IPC in detecting inhibition and interpreting results in ddPCR.

## ACKNOWLEDGEMENTS

We are very grateful to Zhengxin Sun and Claude Lachance for their constructive suggestions and insights on eDNA lab work and results. We thank the Queen's University Biological Station for access to the sampling site. We thank Jacqueline Monaghan for

access to the qPCR instrument for preliminary testing. We are very grateful to James Fotheringham for his support.

### Funding
This work was supported by a NSERC Discovery and a SSHRC New Frontiers in Research Fund grants. The funders had no role in study design, data collection and analysis, decision to publish, or preparation of the manuscript.

### Grant Disclosures
The following grant information was disclosed by the authors:
NSERC Discovery and a SSHRC New Frontiers in Research Fund.

### Competing Interests
The authors declare that they have no competing interests.

### Author Contributions
- Ying Chen conceived and designed the experiments, performed the experiments, analyzed the data, prepared figures and/or tables, authored or reviewed drafts of the article, and approved the final draft.
- Orianne Tournayre conceived and designed the experiments, performed the experiments, analyzed the data, prepared figures and/or tables, authored or reviewed drafts of the article, and approved the final draft.
- Haolun Tian performed the experiments, analyzed the data, prepared figures and/or tables, authored or reviewed drafts of the article, and approved the final draft.
- Stephen C. Lougheed conceived and designed the experiments, authored or reviewed drafts of the article, and approved the final draft.

### Data Availability
The Sanger sequences and their chromatograms are available at Figshare: Tournayre, Orianne (2022): Data from the manuscript "Breeding phenology of a threatened frog species using eDNA and automatic acoustic monitoring". figshare. Dataset. https://doi.org/10.6084/m9.figshare.21112381.v2.

The sequences are also available in the Supplemental Files and GenBank:
- *Pseudacris maculata*: EF988161, KJ536217, KJ536215
- *Pseudacris triseriata*: KJ536225, KJ536224
- *Pseudacris crucifer*: AF488345, AF488344, AF488343, AF488342, AF488341, AF488340, AF488339, AF488338, AF488337, AF488336, AF488335, AF488334, AF488333, AF488332, AF488331, AF488330, AF488329, AF488328, AF488327, AF488326
- *Dryophytes versicolor*: AY831021, AY831016, AY831014, AY831010, AY831009, AY831008, AY831007, AY831006, AY831005, AY831004, AY831003, AY830998,

AY830995, AY830994, AY830992, AY830987, AY830985, AY830981, AY830974, AY830973

   - *Lithobates sylvaticus*: MG002398, MG002397, MG002396, MG002395, MG002394, MG002393, MG002392, MG002391, EU203546, EU203545, EU203544, EU203543, EU203542, EU203541, EU203540, EU203539, EU203538, EU203537, EU203536, EU203535

   - *Lithobates septentrionalis*: AY083273, AY083272, KX269314

   - *Lithobates catesbeianus* KX344492, KX344491, KX344490, KX344489, KX344488, KX344487, KX344486, KX344485, DQ474180, AY210399, AY210398, AY210397, AY210396, AY210395, AY210394, AY210393, AY210392, AY210391, AY210390

   - *Lithobates clamitans*: DQ792700, DQ792699, DQ792698, DQ792697, DQ792696, DQ792695, DQ792694, DQ792693, DQ792692, DQ792691, DQ792690, DQ792689, DQ792688, DQ792687, DQ792686, DQ792685, DQ792684, DQ792683, DQ792682, DQ792681

   - *Lithobates palustris*: KX269353

   - *Lithobates pipiens*: EU370726, EU370725, EU370724, KM396244, KM396243, KM396242, KM396236, KM396235, KM396234, KM396233, KM396232, KM396231, KM396230, KM396229, KM396228

   - *Anaxyrus americanus*: AB159264

## Supplemental Information

Supplemental information for this article can be found online at http://dx.doi.org/10.7717/peerj.14679#supplemental-information.

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

# PeerJ

Pearce-Higgins, Høye TT, Kruuk LEB, Pemberton JM, Sparks TH, Thompson PM, White I, Winfield IJ, Wanless S. 2016. Phenological sensitivity to climate across taxa and trophic levels. *Nature* **535**:241–245 DOI 10.1038/nature18608.

Thalinger B, Deiner K, Harper LR, Rees HC, Blackman RC, Sint D, Traugott M, Goldberg CS, Bruce K. 2021. A validation scale to determine the readiness of environmental DNA assays for routine species monitoring. *Environmental DNA* **3(4)**:823–836 DOI 10.1002/edn3.189.

Thalinger B, Wolf E, Traugott M, Wanzenböck J. 2019. Monitoring spawning migrations of potamodromous fish species via eDNA. *Scientific Reports* **9**:15388 DOI 10.1038/s41598-019-51398-0.

Todd BD, Scott DE, Peachmann JHK, Gibbons JW. 2011. Climate change correlates with rapid delays and advancements in reproductive timing in an amphibian community. *Proceedings of the Royal Society B-Biological Sciences* **278(1715)**:2191–2197 DOI 10.1098/rspb.2010.1768.

Tsuji S, Shibata N. 2020. Identifying spawning events in fish by observing a spike in environmental DNA concentration after spawning. *Environmental DNA* **3(1)**:190–199 DOI 10.1002/edn3.153.

Turner CR, Miller DJ, Coyne KJ, Corush J. 2014. Improved methods for capture, extraction and quantitative assay of environmental DNA from Asian Bigheaded Carp (*Hypophthalmichthys* spp.). *PLOS ONE* **9(12)**:e114329 DOI 10.1371/journal.pone.0114329.

Ulloa JS, Aubin T, Llusia D, Courtois É.A, Fouquet A, Gaucher P, Pavoine S, Sueur J. 2019. Explosive breeding in tropical anurans: Environmental triggers, community composition and acoustic structure. *BMC Ecology* **19(1)**:28 DOI 10.1186/s12898-019-0243-y.

Van Buskirk J, Mulvihill RS, Leberman RC. 2009. Variable shifts in spring and autumn migration phenology in North American songbirds associated with climate change. *Global Change Biology* **15(3)**:760–771 DOI 10.1111/j.1365-2486.2008.01751.x.

Verhaegen B, Reu KD, Zutter LD, Verstraete K, Heyndrickx M, Coillie EV. 2016. Comparison of droplet digital PCR and qPCR for the quantification of shiga toxin-producing *Escherichia coli* in bovine feces. *Toxins* **8(5)**:157 DOI 10.3390/toxins8050157.

Visser ME, te Marvelde L, Lof ME. 2012. Adaptive phenological mismatches of birds and their food in a warming world. *Journal of Ornithology* **153(1)**:75–84 DOI 10.1007/s10336-011-0770-6.

Wake DB, Vredenburg VT. 2008. Are we in the midst of the sixth mass extinction? A view from the world of amphibians. *Proceedings of the National Academy of Sciences of the United States of America* **105(supplement_1)**:11466–11473 DOI 10.1073/pnas.0801921105.

Walther G-R. 2010. Community and ecosystem responses to recent climate change. *Philosophical Transactions of the Royal Society B* **365(1549)**:2019–2024 DOI 10.1098/rstb.2010.0021.

Walther G-R, Post E, Convey P, Menzel A, Parmesan C, Beebee TJC, Fromentin J-M, Hoegh-Guldberg O, Bairlein F. 2002. Ecological responses to recent climate change. *Nature* **416**:389–395 DOI 10.1038/416389a.

While GM, Uller T. 2014. Quo vadis amphibia? Global warming and breeding phenology in frogs, toads, and salamanders. *Ecography* **37(10)**:921–929 DOI 10.1111/ecog.00521.

Whitaker JO Jr. 1971. A study of the western chorus frog, *Pseudacris triseriata*, in Vigo County, Indiana. *Journal of Herpetology* **5(3/4)**:127–150 DOI 10.2307/1562735.

Wilcox TM, McKelvey KS, Young MK, Jane SF, Lowe WH, Whiteley AR, Schwartz MK. 2013. Robust detection of rare species using environmental DNA: The importance of primer specificity. *PLOS ONE* **8(3)**:e59520 DOI 10.1371/journal.pone.0059520.

Zhao Y, Xia Q, Yin Y, Wang Z. 2016. Comparison of droplet digital PCR and quantitative PCR assays for quantitative detection of *Xanthomonas citri* subsp. citri. *PLOS ONE* **11(7)**:e0159004 DOI 10.1371/journal.pone.0159004.