# Peer review of "Assessing the breeding phenology of a threatened frog species using eDNA and automatic acoustic monitoring"

_PeerJ, doi:10.7717/peerj.14679_

## Round 0.1 · original submission · Major Revisions

· Academic Editor

Major Revisions

Dear Authors,

I have received three independent reviews of your study. While all reviewers clearly recognized the quality/novelty of your work, they have collectively raised a number of major issues that will need to be addressed in your revised manuscript.

I'd like to thank all reviewers for their time and effort in improving this study.

Reviewer #2 found the study to be clear & well organized, but noted that parts of the manuscript lacked clarity and provided a list of technical questions for you to address. Reviewer #3 insisted that English and sentences structure needed considerable improvement, while also highlighted issues around PCR inhibition and interpretation/conclusion.

Overall, the reviewers have provided you with excellent suggestions on how to improve the manuscript, and I be looking forward to receiving your revised manuscript along with a point-by-point response to their comments.

With warm regards,
Xavier

Reviewer 1 ·

Basic reporting

I found this article very pleasant to read overall, with rather well written sentences, while remaining very scientific. The introduction clearly sets out the context of the study and its challenges (development of an interesting alternative for the observation of these amphibians, well known to be sensitive species, highly threatened). The objectives are clearly stated and the methodology subsequently developed makes it possible to meet them perfectly.

Experimental design

The study has a real scientific interest and every precaution has been taken to anticipate the slightest flaw in the protocol. As the authors mentionn, they have reached the maximum level of validation of the Thalinger scale, with a very robust protocol in terms of validation:
- in-silico (many mismatches with co-occurring or genetically close species)
- in-vitro (with ddPCR tests on co-occurring and genetically close species)
- in-situ (on sites where the presence of the target species is known, via other studies of the same type)
- the amplification products were sequenced to be sure that the species detected was the correct one.

The method is very well explained, clearly, and can be reproduced easily in future studies.

Validity of the findings

Following this very robust protocol, the results are very robust. The authors even took into account the possibility of having inhibition in their samples (a frequent occurrence in environmental DNA). They therefore tested this potential inhibition, as well as the possibilities (3 tests) to reduce this factor in their samples.
In their results, they took environmental factors seriously and tested their influence on amphibian behavior.

All the results are very robust and have been put forward in a clear way, meeting the objectives stated earlier in the introduction.

Reviewer 2 ·

Basic reporting

Please see the "Additional comments" section

Experimental design

Please see the "Additional comments" section

Validity of the findings

Please see the "Additional comments" section

Additional comments

This article investigates the breeding phenology of the threatened frog species P. maculata using eDNA and automatic acoustic monitoring. The manuscript is clear and well organized, the language is professional, and the context is sufficiently detailed. Additional information is provided by the authors in supplementary materials. Overall, the figures are clear and the legend appropriate.
Some parts of the manuscript lack clarity or could still be improved with additional details. Please find a breakdown of various comment and suggestions regarding various points of the manuscript.
Introduction:
The intro is clear and well detailed; however, it would gain to have a couple of sentences and references related to the use of ddPCR technology (and why this method was chosen instead of qPCR or metabarcoding).
Methods:
Line 128: you started to record water temperature on 8th March. When did you stopped? After July 2022?
Lines 149-160: Did you considered the use of occupancy modelling to assess the impact of false positive/negative or investigate the effect of various parameters collected in the detectability eDNA?
Lines 164-165: can you details which parameters values?
Line 181: This is the first mention of sanger sequencing. How was it done? Did you ran another PCR for sanger sequencing following the ddPCR analysis ?
Line 186: same comment
Line 225: the final elution in a 50ul volume could also explain the high inhibition in some cases. Do you think it would help to elute the DNA in a larger volume (200ul or other)? Or would it decrease the probability of detection?
Line 229-233: did you pooled together the DNA extract of all water samples for a single location? If yes, do you think it increased the inhibition levels?
Line 249: maybe replace "tackling" by "mitigation strategy" (or similar)?
Line 250-262: It is indeed good practice to include IPC. Do you think it would be better to spike the samples before the filtration? This would first allow to know how much DNA was retrieved after extraction, and second assess the inhibition levels.
Results:
Line 300: The LOD is really low there. How many droplets did you obtained in each replicates? Did you pooled the quantification of all replicates together and provided the mean result? LOD at 0.009 copies/uL seems really stochastic detection.
Line 329-332: Do you have a hypothesis why these five samples show higher inhibition?
Line 336-339: Same question there: why the inhibition is lower? Could it be due to the thawing process and release of humic substance and PCR inhibitors?
Discussion
Line 376: It is not clear in the methods that all positive eDNA samples were sanger sequenced. The manuscript will be improved with additional details about this.
You detected eDNA signals after confirming that the species is present after the traditional survey with acoustic monitoring. Do you think that eDNA survey is still beneficial there compared to the already established monitoring? Do you think airborne eDNA detection would allow an earlier detection that acoustic monitoring (this type of sample should at least be less sensitive to inhibition)?
Line 442-444: Do you have any hypothesis to explain this? Ice thawing could potential release humic substance or dilute inhibition effects?

Figures
Figure 6: why is the threshold different between D, E and F? This should be consistent across the student to obtain standardized results.

·

Basic reporting

In general, the text is well-written, but there are some parts where English can be improved. Also, the text seems to be too descriptive and lacks clear structuring in function of the research questions. It sometimes seems like the text is build-up with individual sentence without a clear coherence or link between individual sentences. I made plenty suggestions to increase this aspect of their text. Even though, this is nothing that cannot be solved, I would strongly encourage the authors to increase the English throughout their text.

In terms of the provided background information serving as a context fot he study, I think the authors did a good job in general . However, in my view, the authors elaborated too much on molecules causing PCR inhibition in lentic systems in their discussion section, after which they state"… but we did not investigate this". I would suggest reducing the number of sentences in the discussion covering potential mechanisms of ddPCR inhibitions, to keep the attention of your readers to the scope of this article.

Also, the structure of the text can be improved. The researchers suddenly introduce an additional goal midway in their manuscript, to test methods to reduce PCR-inhibition. I would suggest explicitly adding this to your research goals, by stating that these kind of water samples typically result in PCR-inhibition.

Figures are clear and visually appealing.

Experimental design

In general, their experimental design is very solid and fits the scope of the journal and the special issue. The only exception is the part regarding testing the efficacy of methods mitigating PCR inhibition. Only a very limited number of samples were used, and the results were subjectively described, in stead of tested statistically. I would encourage the authors to invest some additional attention to this part of the study.

Also, the authors suggest that their eDNA concentrations might reflect calling assemblage size. In my view, their experimental setup does not allow this to be investigated, and I would suggest focusing this paper on detection rather than quantification of abundance.

Validity of the findings

Their findings are robust and statistically sound, with the only exception being the part concerning PCR inhibition mitigation methods. However, I am highly concerned about the way the authors dealt with the fluoresence threshold in ddPCR reactions. To the best of my knowledge, this fluorescence threshold is set based on the amplitude of positive and negative droplets retrieved in positive control samples. However, the researchers apparently modified this threshold on a sample-specific basis. I would like the authors to elaborate on this part of their research, and if needed, change the way they set this threshold.

I also have a problem with one of the conclusions of this research ("our results showed the promise of environmental DNA ddPCR as an effective, non-invasive tool for assessing breeding phenology in cryptic amphibians with moderate sampling efforts"), for several reasons:
- the authors detected the eDNA of this frog only 6 days after the first call was detected with acoustic monitoring, suggesting that it is more accurate to assess the phenology of this frog species based on acoustic data in stead of eDNA data.
- The researchers found that their eDNA data was extremely spatially heterogenous, while acoustic detectors covered a much larger spatial area. As a result, the presence of this species is more likely to be detected acoustically than when using eDNA.
- acoustic monitoring is passive and cheap, while eDNA monitoring is expensive and intensive. I think that when a conservation manager want to assess the breeding phenology of a particular species, he/she would prefer installing a few acoustic detectors rather than sampling eDNA
Nonetheless, this work is invaluable as the authors developed and rigorously validated a new eDNA assay that can discriminate between mitotypes, and is actually quite sensitive when compared to the acoustic data. A more general conclusion might indeed be that for other amphibian species that do not call like the one studied here, eDNA might indeed be a good tool to assess its phenology, but I think their main conclusion is that they developed a very robust and sensitive eDNA assay, that is able to accurately detect a specific mitotype of the chorus frog.

Additional comments

In spite of the suggested modificiations, I think this work is a really valuable addition to eDNA research. I think it is really interesting coupling eDNA-based sampling to acoustic monitoring and other climatic variables. I therefore did my best to provide suggestions on how to improve this manuscript (see attached doc), and I would be very happy to see this work published.

---

## Round 0.2 · accepted · Accept

· Academic Editor

Accept

Dear Authors,

I am pleased to accept this revised manuscript for publication in PeerJ - Congratulations!

I also take the opportunity to thank the reviewers for their valuable contribution in improving this work.

This study will represent a great contribution to the field of eDNA research - thank you!

With warm regards,
Xavier

Reviewer 1 ·

Basic reporting

No comment

Experimental design

No comment

Validity of the findings

No comment

Additional comments

For this second round of review, I again accept the manuscript in its current form.
I was already in favor of a publication during the first review phase, the study already seemed to me very robust and interesting, with a well-calibrated protocol according to the constraints of the environment, highlighting a good knowledge of it.
Nevertheless, the other reviewers had interesting and relevant comments and the authors took care to respond seriously and conscientiously to each of them, justifying each response with constructed and documented arguments. In my opinion, this has had the effect of further improving the quality of this manuscript, which now deserves to be published in the journal.

Reviewer 2 ·

Basic reporting

The authors provided a revision of their manuscript following and a point by point response letter. The manuscript is now clearer, and all the authors did a great job to improve the different sections of the manuscript. I'm happy to recommend this manuscript for publication in PeerJ.

Experimental design

see above

Validity of the findings

see above

Additional comments

see above

·

Basic reporting

See additional comments

Experimental design

See additional comments

Validity of the findings

See additional comments

Additional comments

The authors reviewed their work thoroughly, and I am very pleased with the overal result. In my view, this work can now be published in PeerJ.

A few minor comments:
- I am still not convinced about the quantitative aspect of this study. I think the set-up and the resulting data do not allow making inferences about the link between species abundance and eDNA concentration, and therefore, that this is the weak part of this study. However, I appreciate that the authors toned down the claims in the earlier version of the mansucript. Since it seems that other reviewers had no issue with this aspect, I accept the current version of the manuscript.
- There are still a few typo's present in the current version of the manuscript.
- I suggested a relevant reference for their paper, and even though the authors mention that they included this reference, this was not the case. It concerns this reference: https://doi.org/10.1002/edn3.301

I am looking forward to see this nice work published!